# α/β coiled coils

**Marcus D Hartmann, Claudia T Mendler[†], Jens Bassler, Ioanna Karamichali, Oswin Ridderbusch[‡], Andrei N Lupas\*, Birte Hernandez Alvarez\***

Department of Protein Evolution, Max Planck Institute for Developmental Biology, Tübingen, Germany

**Abstract** Coiled coils are the best-understood protein fold, as their backbone structure can uniquely be described by parametric equations. This level of understanding has allowed their manipulation in unprecedented detail. They do not seem a likely source of surprises, yet we describe here the unexpected formation of a new type of fiber by the simple insertion of two or six residues into the underlying heptad repeat of a parallel, trimeric coiled coil. These insertions strain the supercoil to the breaking point, causing the local formation of short $\beta$-strands, which move the path of the chain by 120° around the trimer axis. The result is an $\alpha/\beta$ coiled coil, which retains only one backbone hydrogen bond per repeat unit from the parent coiled coil. Our results show that a substantially novel backbone structure is possible within the allowed regions of the Ramachandran space with only minor mutations to a known fold.

**\*For correspondence:** andrei.
lupas@tuebingen.mpg.de (ANL);
birte.hernandez@tuebingen.mpg.
de (BHA)

**Present address:**
[†]Nuklearmedizinische Klinik und
Poliklinik, Klinikum rechts der
Isar, Technische Universität
München, Munich, Germany;
[‡]Vossius and Partner,
Siebertstraße, Germany

**Competing interests:** The
authors declare that no
competing interests exist.

**Reviewing editor:** Mingjie
Zhang, Hong Kong University of
Science and Technology, China

## Introduction

α-Helical coiled coils are ubiquitous protein domains, found in a wide range of structural and functional contexts (*Lupas, 1996*). They were the first protein fold described in atomic detail (*Crick, 1953b*) and are also the only one whose backbone structure can be computed with parametric equations (*Crick, 1953a*), placing them at the forefront of protein design efforts (*Huang et al., 2014*; *Joh et al., 2014*; *Thomson et al., 2014*; *Woolfson, 2005*).

The structure of coiled coils is understood at a level unrivaled by any other fold. They consist of at least two α-helices, wound into superhelical bundles and held together by a mostly hydrophobic core. In their most prevalent form they follow a heptad sequence repeat pattern. The seven positions in a heptad are labeled *a* – *g*, where positions *a* and *d* are oriented towards the core of the bundle and are thus mostly hydrophobic. Beyond the heptad repeat, a range of other periodicities is accessible to coiled coils, which is only restrained by the periodicity of the unperturbed α-helix (*Gruber and Lupas, 2003*). This restraint is responsible for the supercoiling of the bundle: As an ideal, straight α-helix has a periodicity of about 3.63 residues per turn, the heptad coiled coil has a left-handed twist to reduce the periodicity to 3.5 residues per turn with respect to the bundle axis. In hendecad coiled coils, the situation is reversed: 11 residues are accommodated in 3 helical turns, resulting in 11/3 = 3.67 residues per turn. As this is slightly above 3.63, hendecads are slightly right-handed. With the periodicity of pentadecad coiled coils, 15/4 = 3.75 residues per turn, right-handedness is as pronounced as left-handedness is in heptad coiled coils.

Many naturally occurring coiled coils contain transitions between segments of different periodicity (*Alvarez et al., 2010*; *Hartmann et al., 2014*) or harbor discontinuities that retain the α-helical structure, but perturb the periodicity locally (*Parry, 2014*). The best understood discontinuities are insertions of 3 or 4 residues, which are close to the periodicity of 3.63 of α-helices (*Brown et al., 1996*; *Hicks et al., 2002*; *Lupas and Gruber, 2005*). The insertion of 3 residues is termed a stammer, the insertion of 4 residues a stutter. With 3 residues being less than one full turn of a helix, stammers lead to a local decrease in periodicity and an increase of left-handedness. Stutters have the opposite effect. Inserted into a heptad coiled coil, a stutter can locally extend one heptad to form a hendecad

**eLife digest** Proteins are made up of building blocks called amino acids. Groups of amino acids within the protein can then fold into three-dimensional shapes, one of the most common being a helical structure known as an α-helix. Two or more α-helices may be wound around each other to form a bundle called a coiled coil, which is found in many proteins. Each complete turn of an α-helix contains a set number of amino acids, but the number of amino acids in the turns of a coiled coil can vary. The most common pattern in a coiled coil has 7 amino acids over two turns, which is known as a heptad repeat.

When amino acids are added into or deleted from the heptad repeats, the number of amino acids in the turns of a coiled coil changes. However, it cannot increase too far beyond the number of amino acids in each turn of a normal α-helix because there is a limit to the amount of coiling that the helices can tolerate. Many naturally occurring coiled coils have regions where the overall α-helical structure is retained, even though there are small sections where the number of amino acids in a turn is disrupted. This may be due to insertions of small numbers of amino acids. Although the impact of some insertions (e.g. three or four at a time) has been studied, the effect of inserting other amounts of amino acids was not clear.

Hartmann et al. investigated what would happen when two or six amino acids were inserted into the heptad repeats of a coiled coil within a protein from bacteria. These numbers of amino acids have been predicted to cause the greatest strain on the coiled coil structure. The experiments show that inserting these numbers of amino acids caused so much strain that the three α-helices making up the coiled coil break apart and refold into a completely different type of structure called a β-strand. The three short β-strands then associate into a triangular structure that Hartmann et al. named a β-layer.

Further experiments showed that inserting the same numbers of amino acids into the heptad repeats of other coiled coil proteins also resulted in the formation of β-layers. Hartmann et al.'s findings suggest that the alternating α-helix and β-strand structures may help to make the proteins stronger and enable to carry out more versatile roles in cells.

(7 + 4 = 11 -> 11/3) or, being delocalized over multiple heptads, lead to even higher periodicities like 18 residues over 5 turns (7 + 7 + 4 = 18 -> 18/5). Other periodicities can be brought about by the insertion of multiple stammers or stutters (e.g. 7 + 4 + 4 = 15 -> 15/4). These relationships are illustrated in *Figure 1*, which shows the effects on coiled-coil periodicity resulting from consecutive insertions of stammers (blue lines) and stutters (green lines), and from their progressive delocalization (red lines).

However, there are limits to the periodicities coiled coils can assume, imposed by the degree of supercoiling the constituent helices can tolerate. The insertion of a stammer into a heptad coiled coil, leading locally to a periodicity of 10/3 = 3.33, was predicted to cause an overwinding of the helices (*Brown et al., 1996*). We could verify this experimentally: the structure of a stammer showed that the local overwinding introduced sufficient strain to cause the formation of a short $3_{10}$-helical segment (*Hartmann et al., 2009*). We therefore assume that 3.33 (10/3) residues per turn mark the lower limit for periodicities. As this is about 0.3 residues per turn less than the periodicity of a perfectly straight helix, one might expect the upper limit at a periodicity of about 3.9. In fact the vast majority of known coiled-coil structures deviating from the heptad repeat have periodicities higher than 3.5 and the most extreme example is found in the trimeric autotransporter YadA, which has a local periodicity of 3.8 (19/5) (*Alvarez et al., 2010*).

In contrast to stammers and stutters, accommodating insertions of 1 or 5 residues is more demanding for the bundle. According to *Figure 1* they have to be delocalized over more than one heptad, as periodicities of 4.0 ((7+1)/2) or 2.66 ((7+1)/3) do not fall into the accessible range, and neither do 2.5 (0+5/2), 4.0 ((7+5)/3) or 3.0 ((7+5)/4). To retain α-helical structure, both insertions of 1 and 5 residues have to be delocalized over at least two heptads, leading to periodicities of 3.75 (15/4) and 3.8 (19/5), respectively. Interestingly, these periodicities can also be brought about by the insertion of 2 (15/4) and 3 (19/5) consecutive stutters. Alternatively, insertions of 1 residue (skip

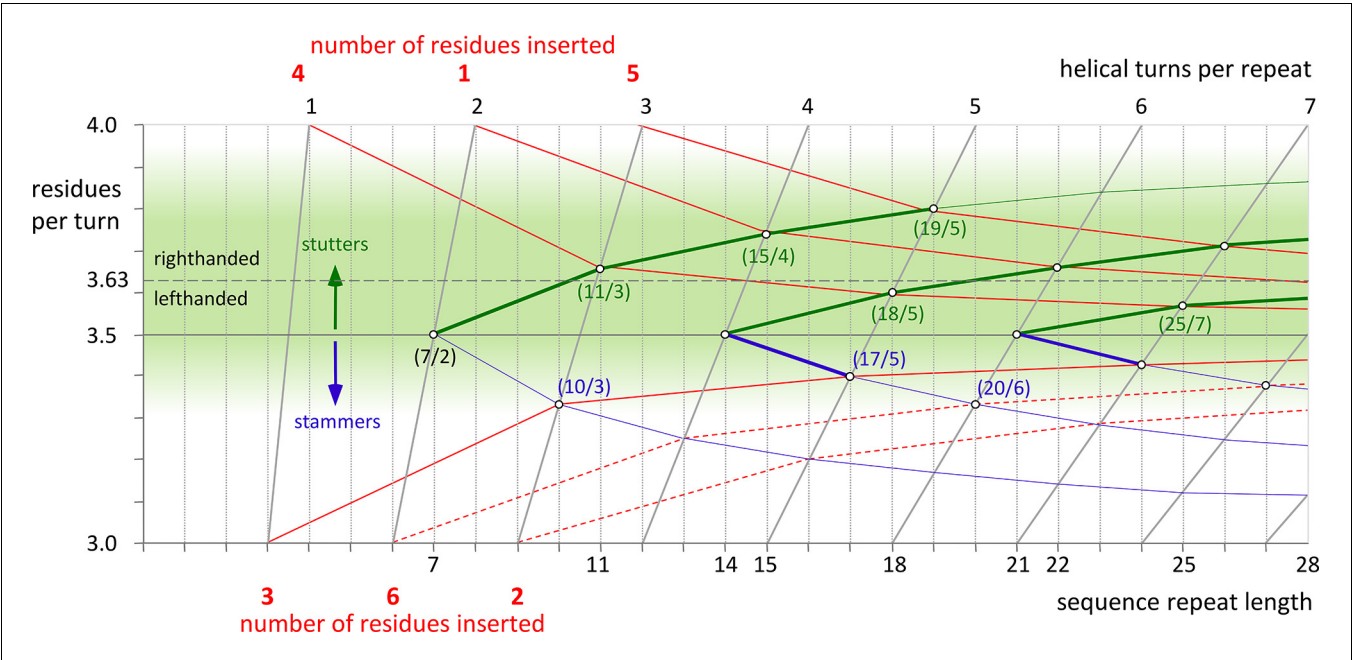

**Figure 1.** Transitions in periodicity caused by insertions of one to six residues into the heptad repeat. The green area marks the estimated boundaries of periodicities accessible to α-helical coiled coils. It is centered around the periodicity of unperturbed α-helices, about 3.63 residues per turn. Higher values than 3.63 lead to right-handed and lower values to left-handed supercoiling. The effects of consecutive insertions of stammers (3 residues) or stutters (4 residues) into a heptad pattern are shown by blue and green lines, respectively. The red lines correspond to the insertion of 1 to 6 residues into the heptad periodicity and their progressive delocalization over neighboring heptads. For example, an insertion of 4 residues is accommodated as 11 residues over 3 turns (11/3), when delocalized over one heptad, or as 18/5, when delocalized over two. Insertions of 1 or 5 residues have to be delocalized over two heptads, resulting in periodicities of 15/4 or 19/5 (which could also be brought about by consecutive stutters – following the green line from 7/2 over 11/3 over 15/4 to 19/5). Insertions of 3 can be accommodated as 10/3, at the very edge of the green area, although in the known examples the α-helices are distorted due to the strong left-handed supercoiling which could be avoided by further delocalization. For insertions of 2 or 6 residues (dashed lines) a strong delocalization would be required to reach the green lawn of accessible periodicities. However, for all constructs in this paper, this is not observed. Via the formation of β-layers these insertions sustain the heptad periodicity as unperturbed as possible.

residues) can be accommodated by the local formation of a π-turn in the α-helix, leaving the remaining coiled coil largely unperturbed (*Lupas, 1996*).

Still missing for a complete picture of coiled-coil periodicities is the understanding of insertions of 2 and 6 residues, which should cause the greatest strain on α-helical geometry. We find that they indeed break the α-helices to form short β-strands, which associate into a triangular supersecondary structure we name the β-layer. β-Layers are found, also repetitively, in natural coiled coils, where they form regular fibers with alternating α- and β-structure, a protein fold that has not been described so far.

## Results and discussion

### A β-layer in the coiled-coil stalk of Actinobacillus OMP100

We have a long-standing interest in trimeric autotransporter adhesins (TAA), fibrous proteins of the Gram-negative bacterial surface (*Bassler et al., 2015*; *Hartmann et al., 2012*; *Hoiczyk et al., 2000*; *Szczesny and Lupas, 2008*), whose domains we routinely fuse to stabilizing adaptor coiled coils for biochemical and biophysical study (*Deiss et al., 2014*; *Hernandez Alvarez et al., 2008*). In the process, we have repeatedly gained insights into aspects of coiled-coil structure (*Alvarez et al., 2010*; *Grin et al., 2014*; *Hartmann et al., 2012*; *2009*; *Leo et al., 2011*), such as for example into a recurrent polar motif of the hydrophobic core (the N@*d* layer), in which asparagines in position *d* of the core coordinate anions at their center (*Hartmann et al., 2009*). As part of that study, we identified a putative TAA in *Actinobacillus actinomycetemcomitans*, OMP100, which carries insertions of 2 and

of 3 residues within the heptad repeats of its stalk. The insertion of 2 residues extends a heptad to the 9-residue motif IENKADKAD and occurs between three N-terminal and two C-terminal heptads carrying N@*d* layers; the insertion of 3 residues is directly downstream. This observation was highly puzzling, since the heptad register of the protein could be assigned with great confidence, leaving no doubt that an insertion of 2 residues had occurred, but this insertion could not be explained by coiled-coil theory. For structural characterization, we therefore expressed residues 133Q-198K, covering the two insertions and the five N@*d* layers, fused N- and C-terminally to the trimeric form of the GCN4 leucine zipper, GCN4-pII (*Table 1*). The construct yielded a typical α-helical CD spectrum and, upon heating, unfolded cooperatively with a transition midpoint at 91°C. We obtained crystals in space group C2 that diffracted to a resolution of 2.3 Å, with one symmetric trimer in the asymmetric unit. The structure showed a continuous heptad coiled coil with two discontinuities (*Figure 2*). As expected, the insertion of 3 residues C-terminal to the N@*d* layers led to the formation of a decad, with a short $3_{10}$-helical segment, as for the stammer we had described previously (*Hartmann et al., 2009*).

However, the insertion of 2 residues led to a sharp break in the coiled coil: In the middle of the IENKADKAD motif, the three chains of the trimer cross each other to form a triangular plane

**Table 1.** Sequences of constructs and protein buffer composition.

| Construct | Protein sequence | Final buffer |
|---|---|---|
| OMP100 | (GCN4-pII)N-**IQNVDVR STENAAR SRANEQK IAENKKA IENKADKAD VEKNRAD IAANSRA IATFRSSSQN IAALTTK**-(GCN4pII)c-KLHHHHHH | 20 mM Tris pH 7.5, 400 mM NaCl, 5% Glycerol |
| Tcar0761 | (GCN4-N16V)N-**ITLMQAN ---MATKDD LARMATKDD IANMATKDD IANMATKDD IAKLDVK IENLNTK**-(GCN4-N16V)c-GSGHHHHHH | 20 mM MOPS pH 7.2, 500 mM NaCl, 5% Glycerol, 2 M Urea |
| T6 | (6xH-TEV)-(GCN4-N16V)N-**MATKDD**-(GCN4-N16V)c | 20 mM HEPES pH 7.4, 50 mM NaCl, 5% Glycerol, 1 M Urea |
| T9 | (6xH-TEV)-(GCN4-N16V)N-**MATKDDIAN**-(GCN4-N16V)c | 20 mM HEPES pH 7.4, 50 mM NaCl, 5% Glycerol, 1 M Urea |
| A6 | (6xH-TEV)-(GCN4-N16V)N-**IENKAD**-(GCN4-N16V)c | 20 mM HEPES pH 7.4, 50 mM NaCl, 5% Glycerol, 1 M Urea |
| A7 | (6xH-TEV)-(GCN4-N16V)N-**IENKKAD**-(GCN4-N16V)c | 20 mM HEPES pH 7.4, 50 mM NaCl, 5% Glycerol, 1 M Urea |
| A9 | (6xH-TEV)-(GCN4-N16V)N-**IENKADKAD**-(GCN4-N16V)c | 20 mM HEPES pH 7.4, 50 mM NaCl, 5% Glycerol, 1 M Urea |
| A9b | (6xH-TEV)-(GCN4-N16V)N-**IANKEDKAD**-(GCN4-N16V)c | 20 mM HEPES pH 7.5, 50 mM NaCl, 10% Glycerol, 1 M Urea |

| | |
|---|---|
| (GCN4-pII)N | MKQIEDKIEEILSKIYHIENEIARIKKL |
| (GCN4-pII)C | MKQIEDKIEEILSKIYHIENEIARIKKLI |
| (GCN4-N16V)N | MKQLEMKVEELLSKVYHLENEVARLKKL |
| (GCN4 N16V)C | MKQLEWKVEELLSKVYHLENEVARLKKLV |
| (6xH-TEV) | MKHHHHHHPMSDYDIPTTENLYFQGH |

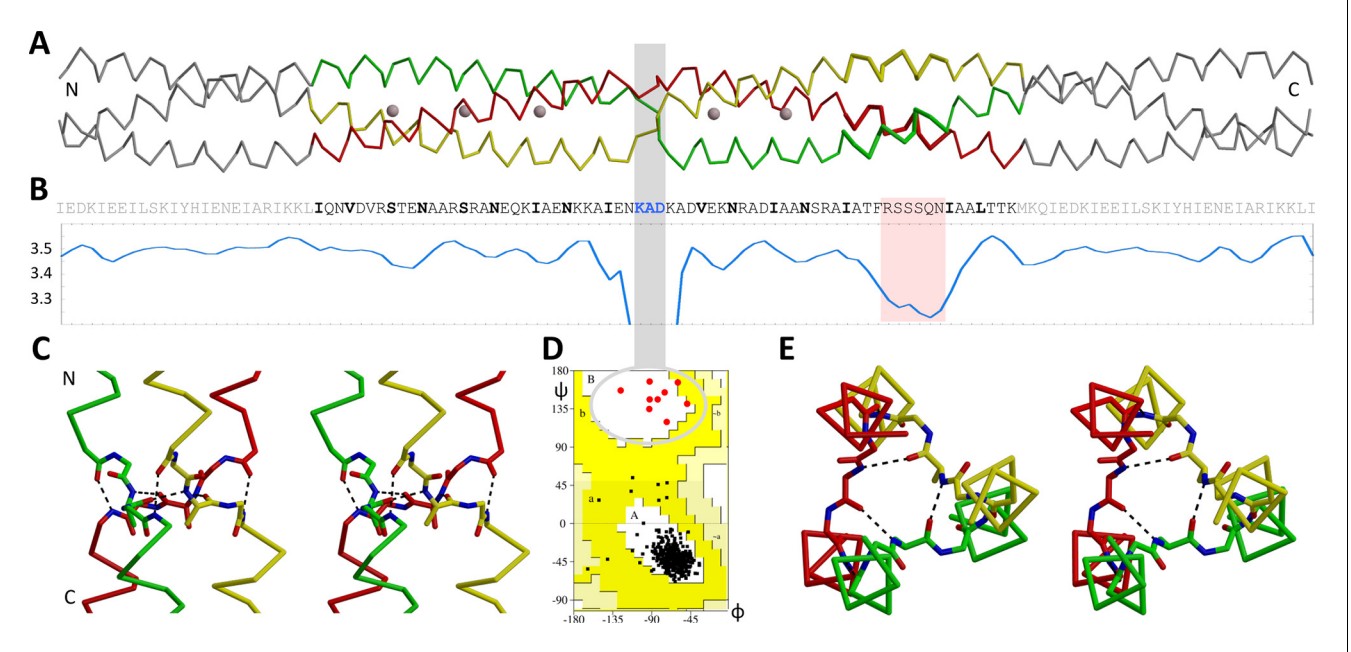

**Figure 2.** The β-layer in the Actionobacillus OMP100 stalk. (**A**) The structure of the Actinobacillus OMP100 stalk construct is aligned with (**B**) its sequence and a periodicity plot. The area of the stammer is highlighted in pink, the three residues of the β-layer by a grey bar. This bar points to the β region of the Ramachandran plot (**D**), where all nine β-layer residues of the trimer are found. The close-ups show the (**C**) side and (**E**) top view in stereo, highlighting the β-layer interactions. The trimer is colored by chain, GCN4 adaptors in grey. The plot is smoothed over a window of three residues to mask local fluctuations. Empty regions of the Ramachandran plot are cropped.

perpendicular to the coiled-coil axis. Thereby only the central three residues of the motif, KAD, deviate from α-helical structure (*Figure 2*). All three fall into the β region of the Ramachandran plot, but only the central residue, alanine, forms backbone hydrogen bonds with the alanine residues of the other chains. We call this structural element a β-layer. It is essentially the same β-layer we described as an adaptor between α-helical and β-stranded segments of TAAs (*Hartmann et al., 2012*). Here it directly connects two α-helical segments, where the C-terminal one is rotated counterclockwise by ~120° around the trimer axis, as viewed from the N-terminus (*Figure 2*).

The first three residues of the IENKADKAD sequence motif occupy heptad positions *a*, *b* and *c* of the N-terminal α-helical segment, the last three residues positions *e*, *f* and *g* of the C-terminal segment. Therefore the β-layer, formed by the three central residues KAD, occurs in place of position *d*. The two segments are stabilized in their relative orientation by backbone hydrogen bonds from the last (*c* position) residue of each N-terminal helix to the first (*e* position) residue in the C-terminal helix of the neighboring chain (*Figure 2C*). This extends the continuous backbone hydrogen-bond network of each α-helix across the chains.

The nature of the discontinuity represented by this β-layer is related to the nature of stammers, but its effects are much stronger. With the insertion of 3 residues, stammers constitute a major strain on the conformation of the constituent helices of the coiled coil. In all examples to date, the resulting overwinding of the helices is absorbed by a short $3_{10}$-helical segment. While these stammers can be best described to be part of a decad, the β-layer in OMP100 occurs in a motif of nine residues, a nonad. As the requirements of a nonad on its helices would be even more extreme than those of a decad, the strategy for its accommodation is a local but complete departure from helical structure.

## β-layers in GCN4 fusions

Given the structural simplicity of β-layers, we wondered whether these could be brought about more generally by insertions of 2 residues into heptad coiled coils. Furthermore we wondered whether insertions of 6 residues, which pose similar demands on the coiled coil (*Figure 1*), also lead to the formation of β-layers. To tackle these questions experimentally, we designed a set of

constructs that had either 6 or 9 residues inserted between two consecutive GCN4 N16V adaptors, based on two different sequence motifs (*Figure 3*, *Table 1*). One motif is IENKADKAD from *Actinobacillus* OMP100. The other, MATKDDIAN, is from a second family of prokaryotic coiled-coil proteins that we found to contain nonads and related periodicities; it occurs for example in 14 consecutive repeats in the protein Tcar0761 of *Thermosinus carboxydivorans*. From *Actinobacillus* OMP100 we derived the constructs A9 with the full IENKADKAD motif and A6 with the shortened motif IENKAD, as well as the 'control' construct A7 with the 7-residue motif IENKKAD. From *Thermosinus* Tcar0761 we derived the constructs T9, with the full MATKDDIAN motif, and T6, with the shortened motif MATKDD. The GCN4 N16V variant can form both dimeric and trimeric coiled coils and was chosen for these constructs to test for the oligomerization specificity of the inserts. All five constructs were resistant to proteolysis by proteinase K, showed typical α-helical CD spectra, did not melt upon heating to 95°C and yielded well-diffracting crystals. The structures of all constructs were trimeric and could be solved by molecular replacement, using the trimeric GCN4 structure as a search model. For T9, two structures were solved in alternative conformations (*Figure 3*). Apart from A7, which carries a heptad insert, all structures formed β-layers. These are identical in their structure (*Figure 4*), although they are not accommodated in the same way.

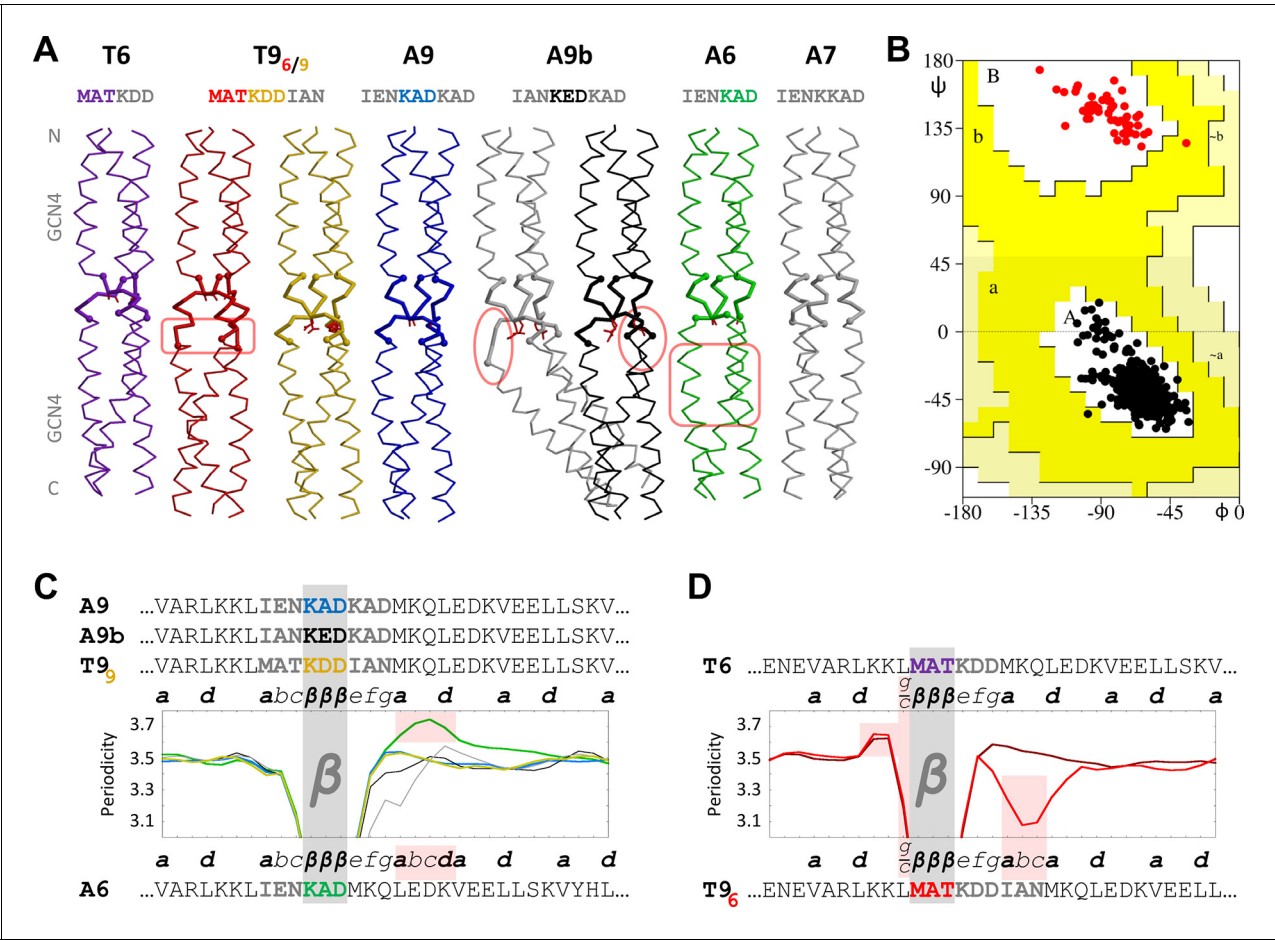

Figure 3. β-layers in 6- and 9-residue motifs between GCN4 adaptors. (**A**) The sequences and structures of the GCN4-fusion constructs are shown together with (**B**) a Ramachandran plot of their backbone torsion angles and (**C, D**) their periodicities. In the structures, the inserts between the GCN4 adaptors are drawn with thick lines. Disturbances in the α-helical segments are highlighted in pink; the stutter in the A6 structure and the stammer in the T9₆ structure are also highlighted in pink in panels C and D. In the periodicity plots, all proteins are aligned on the β-layer and their coiled-coil registers are indicated. The plots are shown separately for β-layers forming nonads (C) and hexads (D). A glitch in the periodicity caused by the *g/c* position preceding β-layers in hexads is highlighted in pink in panel D. As in the previous figure, the periodicity plots are smoothed over a three-residue sliding window. The Ramachandran plot in panel B includes all structures except the kinked grey A9b structure; all residues of the β-layers are shown as red dots and all other residues as black dots. Again, empty regions of the Ramachandran plot are cropped.

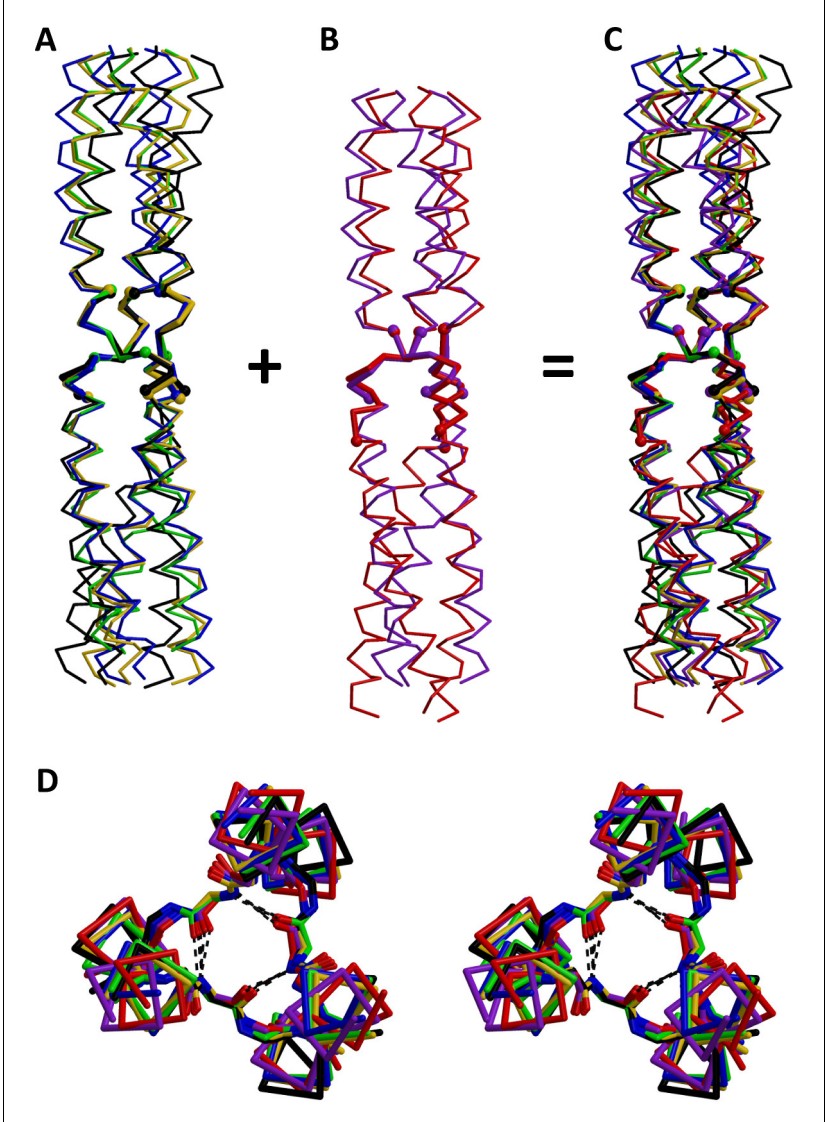

**Figure 4.** Superimposition of β-layers. All structures of β-layers between GCN4 adaptors were superimposed on the actual β-layer elements. Superimpositions are shown separately for β-layers occurring (**A**) in nonads (T9$_9$, A9, A9b, A6) and (**B**) in hexads (T6, T9$_6$); the kinked A9b structure (grey in *Figure 3*) is omitted. Panel (**C**) shows all β-layers together. (**D**) Stereo view of the β-layer region in panel C, seen from the N-terminus. The structures are colored as in *Figure 3*.

In A9 and in one of the T9 structures, T9$_9$, the β-layers are formed as in *Actinobacillus* OMP100. The first three and last three residues of the insert are in heptad register with the flanking GCN4 adaptors and therefore constitute positions *a, b, c* and *e, f, g*. The middle three residues, KAD in A9, KDD in T9$_9$, form the β-layer in place of position *d* (*Figure 3C*). The A6 structure follows the same principle: again the first three residues of the insert are in heptad register with the N-terminal GCN4-adaptor, constituting positions *a, b* and *c*. The other three residues of the insert, KAD, form the β-layer in place of position *d*. As a consequence, the register of the C-terminal GCN4-adaptor is shifted at the junction to the insert, starting with position *e* instead of position *a*. This register conflict is resolved further downstream by the formation of a hendecad (highlighted in pink in *Figure 3A and C*), so that the second half of the C-terminal adaptor retains its original register. In essence, the A6 structure shows a 9-residue element with the same structure as those found in A9

and $T9_9$, where the sequence IENKADMKQ borrows the last three residues from the C-terminal CGN4 adaptor and changes the periodicity of the latter at the junction.

In contrast, the T6 structure shows a 'real' 6-residue element. Here, the $\beta$-layer is formed by the first three residues of the insert, MAT (*Figure 3D*). The last three residues of the insert assume geometrically clear *e, f* and *g* positions and the C-terminal GCN4 adaptor follows in its native register, starting with an *a* position. Therefore, the $\beta$-layer occurs again in place of position *d*. A conflict with the native register of the N-terminal adaptor is avoided with just a small 'twist' to the adaptor's last residue: The C-terminal leucine, natively occurring in position *g*, is rotated outward from the core of the bundle by about 15° so that its Crick angle is biased towards the angle of a *c* position. In *Figure 3* this is noted as a *g/c* position, as it is close enough to an ideal position *g* for the preceding coiled coil to stay in register and close enough to a position *c* for the formation of the subsequent $\beta$-layer (highlighted in pink in *Figure 3D*). Surprisingly, the alternative T9 structure, $T9_6$, starts out in the same way, forming the $\beta$-layer with the first three residues of the insert, MAT, after a *g/c* position. Consequently, the middle three residues constitute positions *e, f* and *g*. It thereby shows the same 6-residue element as the structure T6. The last three residues of the insert are accommodated as a sharply localized stammer, before the C-terminal adaptor starts in position *a* (highlighted in pink in *Figure 3A and D*). Thus, the two structures T6 and $T9_6$ show that 6-residue elements are accommodated as N-terminally shortened 9-residue elements. While $\beta$-layers seem to strictly dictate the downstream register to start with an *e* position, they can occur after both *c* and *g* positions.

The observation that the T9 construct could accommodate the MATKDDIAN insert in two ways, forming the $\beta$-layer either at MAT or at KDD, led us to wonder whether the same could happen with the A9 insert, IENKADKAD, if the glutamate was interchanged with the central alanine to mirror the first six residues of the T9 insert (A9b, IANKEDKAD). We had previously found that $\beta$-layers which occur as connectors between TAA domains prefer small, hydrophobic residues in their central position (*Hartmann et al., 2012*; *Bassler et al., 2015*). We therefore thought that the central aspartate of the T9 insert might have been sufficiently unfavorable ($T9_9$) that an alternative, with the alanine of MAT at the center of the $\beta$-layer ($T9_6$), became observable, even though $T9_9$ allows the flanking coiled-coil segments to remain unperturbed and $T9_6$ requires their distortion. We reasoned that the larger glutamate residue at the center of A9b might even be sufficiently unfavorable to move this construct quantitatively to the alternative structure, with the $\beta$-layer formed over the first three residues (IAN). A9b in fact crystallized in two alternative structures (*Figure 3*), but in both the $\beta$-layer formed over the central glutamate. Since this residue was indeed too large and polar to be accommodated without distortion, the first turns of the downstream helices are perturbed to different extents in both instances, leading to a pronounced kink in one of the structures (highlighted in pink in *Figure 3A*). We were surprised to see that the penalty introduced by the central glutamate was not sufficient to produce the alternative structure; the reasons for this are unclear to us at present.

## The α/β coiled coil

With the expectation to obtain a continuous fiber of alternating α and β elements, we built a construct with repeating nonads, based on *Thermosinus* Tcar0761 (*Figure 5*). The 14 consecutive, almost perfect MATKDDIAN repeats in this protein are flanked by long heptad segments. In our construct we omitted the middle ten nonad repeats and trimmed the N- and C terminal heptad segments for in-register fusion to GCN4-N16V (*Table 1*; red sequence in *Figure 5*). Crystallization trials yielded crystals in space group $P6_3$, diffracting to a resolution of 1.6Å, with one chain in the asymmetric unit and the trimer built by crystallographic symmetry around the c axis. The structure could be solved by molecular replacement using fragments of the T6 and A9 structures. It shows the anticipated α/β coiled coil with four consecutive $\beta$-layers. These layers are formed by the residues MAT of the repeats; the other residues, corresponding to KDDIAN, constitute positions *e,f,g,a,b,c* of the segments between the $\beta$-layers. Therefore, in accordance with heptad notation, the repeats can be written as IANMATKDD, with the isoleucine forming classical hydrophobic *a* layers and the MAT forming $\beta$-layers in place of position *d* (*Figure 5*). Only the first $\beta$-layer is part of a 6-residue element (hexad) and occurs after a position *g* of the preceding heptad. This *g* position is biased towards a *c* position, as described above for the structures T6 and $T9_6$, yielding the same *g/c* position. With its alternating *a*- and $\beta$-layers, the α/β coiled coil is a new class of protein fiber, based on a novel supersecondary structure element.

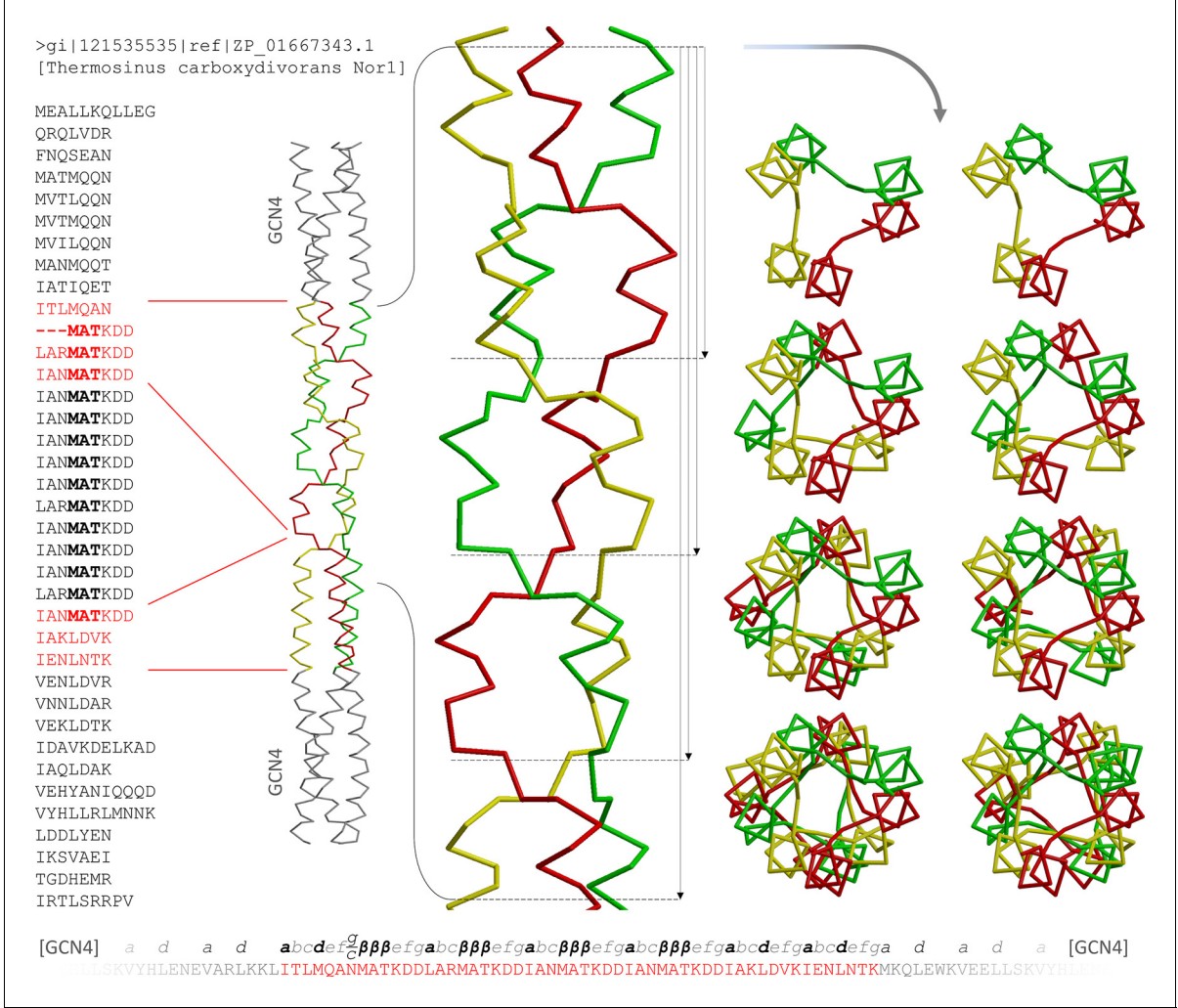

**Figure 5.** The α/β coiled coil in the Tcar0761 construct. The two regions fused between GCN4 adaptors in our construct are shown in red on the full sequence of Tcar0761 (left). Next to the sequence, the structure is depicted as a Cα-trace and the four consecutive β-layers are enlarged. On the right, top views are shown, looking down the bundle from the N-terminus. As indicated by the arrows next to the side view, they show 1, 2, 3 or all 4 β-layers. At the bottom, the sequence of the construct is shown together with the assigned register.

The α/β coiled coil of *Thermosinus* Tcar0761 is built of nonads and thus contains six residues per repeat in the α region of the Ramachandran plot, which retain one backbone hydrogen bond characteristic of α-helical structure. We think that it should be possible to reduce this structure by removing three α-helical residues and thus the single remaining backbone hydrogen bond from the parent structure. Such a minimalistic α/β coiled coil would be built of hexads, with three residues in the β and three in the α region of the Ramachandran plot. We have not so far detected coiled-coil proteins with β-layers in hexad spacing, nor have we been successful in constructing such a structure by fusion of MATKDD repeats between GCN4-N16V adaptors. However, as we will show in the next section, a tail-fiber protein from a *Streptococcus pyogenes* prophage (2C3F) contains an α/β coiled coil with four β-layers, two of which are in a hexad spacing.

## β-Layers in proteins of known structure

At the beginning of this project we had identified nonads in the stalks of TAAs and in the N-terminal coiled coils of a family of prokaryotic endonucleases listed in Pfam as PD-(D/E)XK, specifically in the crenarchaeal representatives of this family. The bacterial representatives, where they had the coiled-coil stalk, lacked nonads or related periodicities (in Pfam however, all the coiled-coil segments of

this family are grouped together in entry DUF3782). Surprisingly, we found that some bacteria contain coiled-coil proteins that lack the endonuclease domain, but are very similar to the coiled coils of the crenarchaeal proteins; *Thermosinus* Tcar0761 belongs to these. The β-layers in this family have the consensus sequence [aliphatic]–A–T–K–[polar]–[DE] (*Figure 6*). Pattern searches with this motif

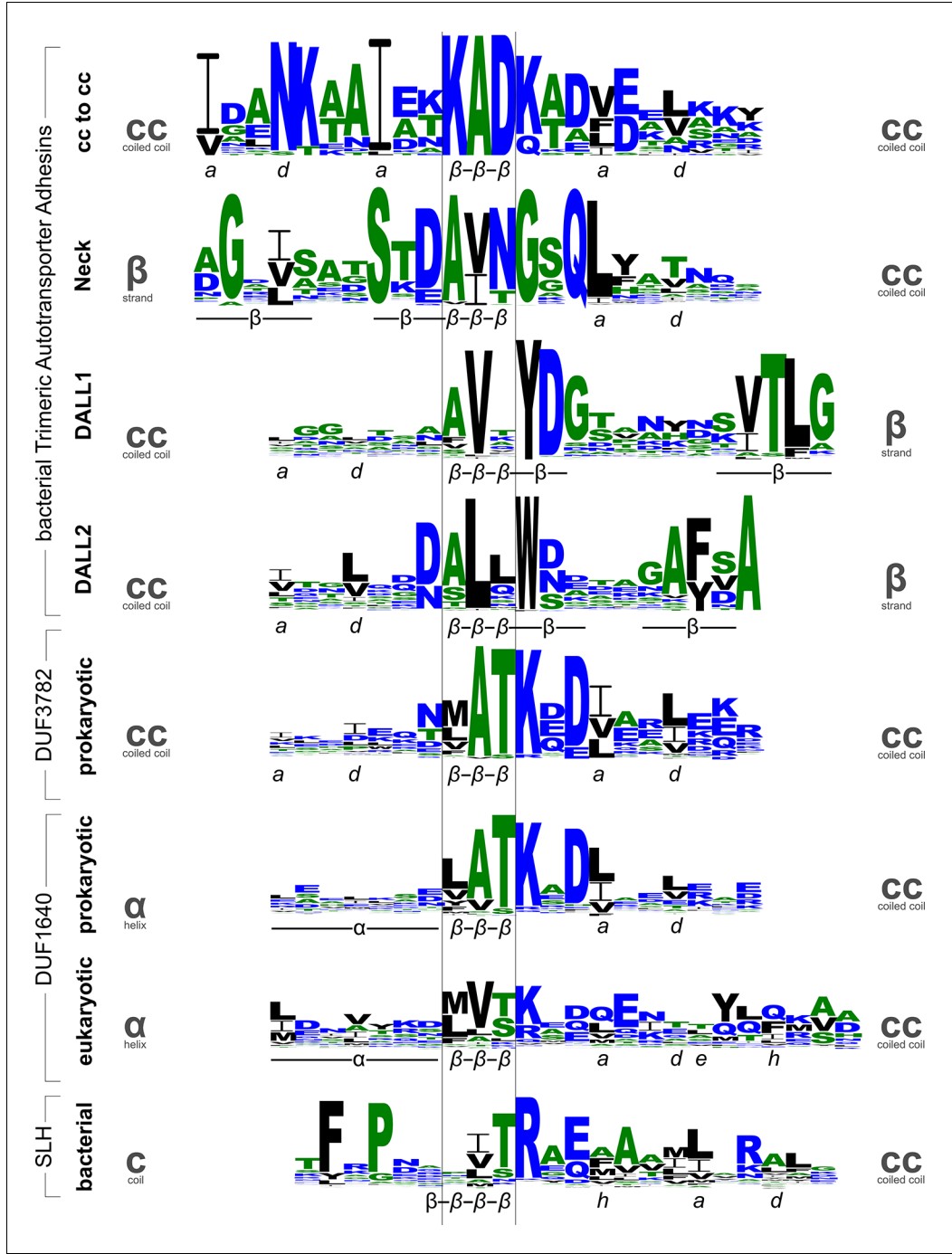

**Figure 6.** Sequence logos for β-layers in different protein families. The sequence logos show the conservation patterns of β-layers and their adjacent secondary structure elements in domains of Trimeric Autotransporter Adhesins (stalk, neck, and two variants of the DALL domain), the DUF3782 family of prokaryotic endonucleases, the DUF1640 family of membrane proteins from prokaryotes and organelles, and the surface layer homology (SLH) domain of bacteria. Annotations of the secondary structure (α: helix, β: strand) and coiled-coil register are shown beneath the logos. Grey symbols on the sides indicate the type of secondary structure transition mediated by the β-layer.

led us to the discovery of a family of integral membrane proteins found in prokaryotes and mitochondria (DUF1640), which carry this motif prominently at the beginning of their C-terminal stalk (*Figure 6*). However, our sequence searches, both based on sequence patterns and on the discovery of relevant insertions into the heptads of coiled coils, have progressed slowly, as they require much case-by-case analysis. This is due on the one hand to the frequency of the $\beta$-layer sequence patterns in coiled coils and on the other to the difficulty of establishing reliably the local register of coiled coils that deviate from the heptad repeat with existing software. Indeed, as we have described for TAAs (*Szczesny and Lupas, 2008*; *Bassler et al., 2015*), many of these escape detection entirely with current programs.

Given the structural identity between the $\beta$-layers resulting from hexads and nonads in coiled coils, and the supersecondary structures we characterized at the transition between coiled-coil segments and β-stranded domains in TAAs (*Hartmann et al., 2012*), we searched systematically for other instances of $\beta$-layers in proteins of known structure. The results are collected in *Figure 7* and *Table 2*. All proteins we identified are homotrimers, except for the SLH domain, which is a monomer with pseudo-threefold symmetry; a majority are from viruses, mainly bacteriophage. Most $\beta$-layers occur in the context of coiled coils and we have termed these 'canonical'. They are usually found capping one of the ends of the coiled coil, more often the N-terminal than the C-terminal one, and we have found only two further examples of coiled coils with internal $\beta$-layers: MPN010, a protein of unknown function from *Mycoplasma pneumoniae* (2BA2), and the aforementioned tail-fiber protein with hyaluronidase activity from the *Streptococcus pyogenes* prophage SF370.1 (2C3F). The latter contains a coiled coil with four $\beta$-layers, one near the N-terminus, two internal, and one at the C-terminal end; the two internal $\beta$-layers have the sequence LQQ**KAD**KET**VYT**KAE and are thus in a hexad spacing, with the first resembling the $\beta$-layer sequence of OMP100 and the second the one of Tcar0761. Remarkably, this second $\beta$-layer deviates from canonical $\beta$-layer structure, which we attribute to the serine in the first core position of the downstream coiled coil. This serine spans a water network in the core of the trimer, which invades the β-interactions of the $\beta$-layer with bridging waters, leading to a largely increased diameter of the layer. This wider diameter might be further promoted by the bulky tyrosine side chain of the central $\beta$-layer residue, which is bent out of the core. Nevertheless, such tandems with the consensus sequence Lxx**KAD**Kxx**VYT**KxE occur in many bacterial ORFs (also in some TAAs, such as *Neisseria meningitidis* NadA4) and thus probably constitute a co-optimized module.

A structural analysis of canonical $\beta$-layers in light of their conserved sequence patterns (*Figure 6*, *Table 2*) shows that they favor hydrophobic residues in $\beta_1$ and $\beta_2$ (*Figure 6*), and particularly the $\beta_2$ residue tends to be of smaller size (i.e. A or V). They can follow upon either position *a* or *d* of the preceding coiled coil, but always lead into positions *e, f, g* of the following coiled coil. Thus, when they follow upon position *a* they yield the register *a-b-c-$\beta_1$-$\beta_2$-$\beta_3$-e-f-g* (seen in nonads), whereas when they follow upon position *d* they bias the residue in position *g* towards *c* to yield the register *e-f-g/c-$\beta_1$-$\beta_2$-$\beta_3$-e-f-g* (seen in hexads). For the purpose of the following discussion we will refer to these two registers collectively as $\alpha_1$-$\alpha_2$-$\alpha_3$-$\beta_1$-$\beta_2$-$\beta_3$-e-f-g.

For $\beta$-layers that occur at the C-terminal end of coiled coils (for example in the DALL1 and DALL2 domains of TAAs), the flanking residues do not form conserved mainchain or sidechain interactions with the layer or with each other, and their conservation pattern is dominated by interactions with the downstream domain. Since $\beta$-layers can form interaction networks that provide a C-cap to the preceding coiled coil (see below), it is surprising that they do not do so in most structures where they occur at the C-terminal end of coiled coils.

For $\beta$-layers that occur at the N-terminal end (for example in the necks of TAAs or in influenza hemagglutinin HA$_2$), the $\beta_3$ residue acts as an N-cap for the following helix, coordinating the backbone NH group of residue *g* (*Figure 8*); it is thus almost always D, N, T, or S (the capping role of this residue has been described in detail in the fusion-pH structure of influenza hemagglutinin HA$_2$ (*Chen et al., 1999*). In return, the sidechain of the residue in position *g* forms a hydrogen bond with the backbone NH group of the $\beta_3$ residue, closing a ring of sidechain-backbone interactions between these two residues; it is thus almost always D, E, or Q. Where it is D or E, it can further form a salt bridge to the residue in position *e* of the neighboring chain (clockwise as viewed from the N-terminus), which is broadly conserved as K or R. This residue essentially always forms either this salt bridge, or a hydrogen bond with the backbone carbonyl group of the $\beta_1$ residue, as depicted in *Figure 8*. This interaction network allows $\beta$-layers to form stably at the N-terminal end of

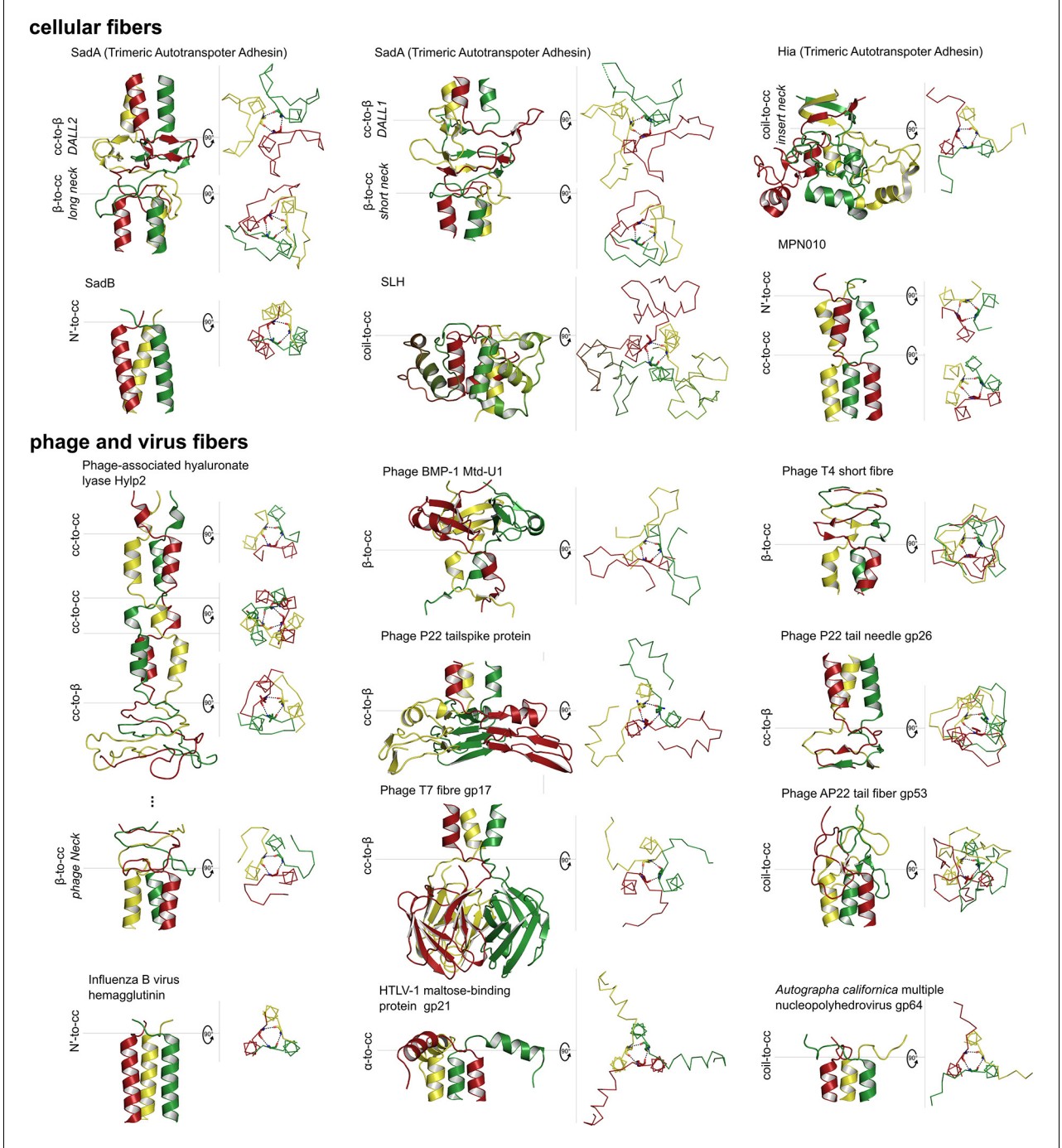

**Figure 7.** Gallery of canonical $\beta$-layers in proteins of known structure. The parts of the structures containing $\beta$-layers are shown in side view (cartoon depiction, left) and the $\beta$-layers in top view (backbone trace, right), with their central ($\beta_2$) residues in stick representation. *Table 2* lists the detailed information for the presented proteins.

coiled-coil proteins, as seen in the crystal structures of SadB, MPN010, and the fusion-pH structure of influenza hemagglutinin $HA_2$.

$\beta$-Layers that occur within coiled coils show substantially the same interactions and conservation patterns as the ones that act as N-caps, when the residue in $\beta_1$ is hydrophobic. Only occasionally, the K or R in position *e* shows yet a third conformation, coordinating the backbone carbonyl of the $\alpha_2$ residue of the preceding helix in the neighboring chain (counterclockwise), thus providing a C-

**Table 2.** β-Layers in proteins of known structure.

**Cellular proteins (canonical)**

| PDB | Type | Protein | Domain | Species | Sequence | Similar structures |
|---|---|---|---|---|---|---|
| 2YO3 | cc-to-β | SadA | TAA DALL1 | Salmonella enterica | abcdefg**βββ**EEEECC 1306-LKASEAGS**VRY**ETNAD-1321 | 3WPA, 3WPO, 3WPP, 3WQA (Acinetobacter sp. Tol5), 4USX (Burkholderia pseudomallei) |
| 2YO2 | cc-to-β | SadA | TAA DALL2 | Salmonella enterica | abcdefg**βββ**EECCC 310-VAGLAED**ALL**WDESI-324 | 3ZMF, 2YNZ (Salmonella enterica) |
| 2YO3 | β-to-cc | SadA | TAA Short neck | Salmonella enterica | EEEECCC**βββ**efgabcdefg**hi**jk**l**mno 1345-AAVNDTD**AVN**YAQLKRSVFEEANTYTDQK-1372 | 4LGO (Bartonella quintana), 3WP8, 3WPA, 3WPR (Acinetobacter sp. Tol5), 1P9H (Yersinia enterocolitica), 2XQH (Escherichia coli), 3D9X (Bartonella henselae), 2YO0 (Salmonella enterica), 3S6L, 4USX (Burkholderia pseudomallei), 2GR7 (Haemophilus influenzae) |
| 2YO2 | β-to-cc | SadA | TAA Long neck | Salmonella enterica | EEEE**βββ**efgabcdefg 349-DSTD**AVN**GSQMKQIEDK-365 | 2YNZ, 3ZMF (Salmonella enterica), 3EMO (Haemophilus influenzae), 3LAA, 3LA9, 4USX (Burkholderia pseudomallei), 3WPA, 3WPO, 3WPP, 3WPR, 3WQA (Acinetobacter sp. Tol5), 3NTN, 3PR7 (Moraxella catarrhalis) |
| 1S7M | β-to-cc | Hia | TAA Insert neck 1 | Haemophilus influenzae | EE**βββ**efgabc 642-NT**AAT**VGDLRG-652 | 3EMF (Haemophilus influenzae) |
| 4C47 | Nterm-to-cc | SadB | - | Salmonella enterica | CC**βββ**efgabcdefg 23-DY**FAD**KHLVEEMKEQ-37 | - |
| 5APP | cc-to-cc | OMP100 | TAA Stalk | Actinobacillus actinomycetemcomitans | abcdefgabc**βββ**efgabcdefg 153-IAENKKAIENKADKADVEKNRAD-175 | - |
| 5APZ | cc-to-cc | Tcar0761 | DUF3782 | Thermosinus carboxydivorans | abcdefg**βββ**efgabc 68-ITLMQAN**MAT**KDDLAR-83 | - |
| 2BA2 | Nterm-to-cc | MPN010 | DUF16 | Mycoplasma pneumoniae | CCC**βββ**efghijk 5-GTR**YVT**HKQLDEK-17 | - |
| 2BA2 | cc-to-cc | MPN010 | DUF16 | Mycoplasma pneumoniae | hijkabc**βββ**efgabcdefghijk 14-LDEKLKN**FVT**KTEFKEFQTVVMES-37 | - |
| 3PYW | coil-to-cc | S-layer protein Sap | SLH | Bacillus anthracis | CCCCE**βββ**efghijkabcdef 35-FEPGK**ELT**RAEAAITMMAQILN-55 . . . 94-FEPNG**KID**RVSMASLLVEAYK-114 . . . 156-WE**PKKTVT**KAEAAQFIAKTDK-176 | - |

**Phage and virus proteins (canonical)**

| PDB | Type | Protein | Domain | Species | Sequence | Similar structures |
|---|---|---|---|---|---|---|
| 2C3F | cc-to-cc | Tail fiber hyaluronidase | - | Streptococcus pyogenes (prophage SF370.1) | abc**βββ**efghijkabc**βββ**efg**βββ**efghijk 69-ID**GLAT**KVETAQKLQQK**AD**KETV**YT**KAESKQE-99 | 2DP5 (Streptococcus pyogenes) |
| 2C3F | cc-to-β | Tail fiber hyaluronidase | - | Streptococcus pyogenes (prophage SF370.1) | defgabc**βββ**CEEEEE 97-SKQELDK**KLN**LKGGVM-112 | 2DP5 (Streptococcus pyogenes) |

*Table 2 continued on next page*

*Table 2 continued*

**Cellular proteins (canonical)**

| PDB | Type | Protein | Domain | Species | Sequence | Similar structures |
|---|---|---|---|---|---|---|
| 2C3F | β-to-cc | Tail fiber hyaluronidase | TAA short neck homolog | *Streptococcus pyogenes* (prophage SF370.1) | EEEECCEβββefghijkabcdefg 310-DPTANDHAATKAYVDKAISELKKL-327 | 2DP5, 2WH7, 2WB3 (*Streptococcus pyogenes*) |
| 4MTM | coil-to-cc | gp53 | - | Bacteriophage AP22 | CCCCβββefgabcdefg 155-NDVGSALSAAQGKVLNDK-172 | - |
| 1YU4 | β-to-cc | Major tropism determinant U1 variant (Mtd-U1) | - | *Bordetella* Phage BMP-1 | CCCEEβββefgab 41-TAGGFPLARHDLVK-54 | - |
| 1TSP | cc-to-β | Tailspike protein | Phage P22-tail | Phage P22 | defghijkβββEEE 113-YSIEADKKFKYSVK-126 | 1CLW, 2XC1, 2VFM, 2VFP, 2VFQ, 2VFO, 2VFN [...] (Phage P22) 4QJP, 4QJ5, 4QJL [...] (*E. coli* Bacteriophage CBA120) 2V5I (Bacteriophage Det7). 2X3H (Enterobacteria phage K1-5) |
| 2POH | cc-to-β | Phage P22 tail needle gp26 | - | Phage P22 | abcdefgβββCEEC 133-ISALQADYVSKTAT-146 | 3C9I, 4LIN, 4ZKP, 4ZKU, 5BU5, 5BU8, 5BVZ (Phage P22) |
| 1H6W | β-to-cc | Short fiber | Receptor binding domain | Bacteriophage T4 | EEEEEECCEEβββefgabcde 321-MTGGYIQGKRVVTQNEIDRTI-341 | 1OCY, 1PDI, 2XGF, 2FKK, 2FL8 (Bacteriophage T4) |
| 4A0T | cc-to-coil | gp17 | gp37_C | Bacteriophage T7 | cdefghijkβββCCCC 454-WLDAYLRDSFVAKSKA-469 | 4A0U (Bacteriophage T7) |
| 1MG1 | α-to-cc | Maltose-binding protein GP21 | TLV_coat | Primate T-lymphotrophic virus 1 (HTLV-1) | HHHHHHβββefgabcdefghijk 364-AAQTNAAAMSIASGKSLLHEVDKD-387 | - |
| 3DUZ | coil-to-cc | GP64 | Baculo_gp64 | *Autographa californica* Multiple Nucleopolyhedrovirus | CCCβββefgabcdefg 293-EGDTATKGDLMHIQEE-308 | - |
| 4NKJ | Nterm-to-cc | Hemagglutinin HA2 | Hemagglutinin HA2 | Influenza B virus | Eβββefgabcdefghijk 4-VAADLKSTQEAINKITKN-21 | 1QU1 (Influenza A virus) |

**Unusual β-layer proteins**

| PDB | Type | Protein | Domain | Species | Sequence | Similar structures |
|---|---|---|---|---|---|---|
| 4NQJ | α-to-α | TRIM Ubiquitin E3 ligase | DUF3583 | *Homo sapiens* | HHHHHHβββHHHHHHH 143-SVGQSKEFLQISDAVHF-159 | - |
| 2F0C | (cc-to-)coil-to-β | Receptor binding protein (ORF49) | - | *Lactophage* tp901-1 | abcdefgabCCCCβββCEEC 22-LEAINSELTSGGNVVHKTGD-41 | 3D8M, 3DA0 (*Lactophage* tp901-1) |
| 1AA0 | (cc-to-)coil-to-β | Fibritin | Fibritin_C | Bacteriophage T4 | abcdefgCβββEEEEE 450-VQALQEAGYIPEAPRD-465 | 1AVY, 2BSG, 2IBL, 2WW6, 2WW7, 3ALM (Bacteriophage T4), 5COR (Influenza A), 2LP7 (Human Immunodeficiency Virus 1), 1NAY |
| 2XGF | coil-to-coil | Long tail fiber needle | - | Bacteriophage T4 | EEEECCCCCCCCβββCCCEEEE 934-EAWNGTGVGGNKMSSYAISYRAG-956 | - |

*Table 2 continued on next page*

*Table 2 continued*

**Cellular proteins (canonical)**

| PDB | Type | Protein | Domain | Species | Sequence | Similar structures |
|-----|------|---------|--------|---------|----------|--------------------|
| 1H6W | coil-to-coil-(to-β) | Short fiber | - | Bacteriophage T4 | CCC***βββ***CCCCEEEEE 284-NAD**VIH**QRGGQTING-298 | - |
| 4UXG | β-to-coil | Proximal long tail fibre protein gp34 | - | Bacteriophage T4 | EEE***βββ***CCCCCC 1233-FVQ**VFD**GGNPPQ-1244 | - |
| 4UXG | α-to-coil | Proximal long tail fibre protein gp34 | - | Bacteriophage T4 | HHHHC***βββ***CCCEEE 1245-PSDIG**ALP**SDNATM-1258 | - |
| 3QC7 | α-to-coil | Head fiber | - | Bacteriophage Phi29 | HHHHHHH***βββ***CCCCCCC 221-NLRTMIG**AGV**PYSLPAA-237 | - |

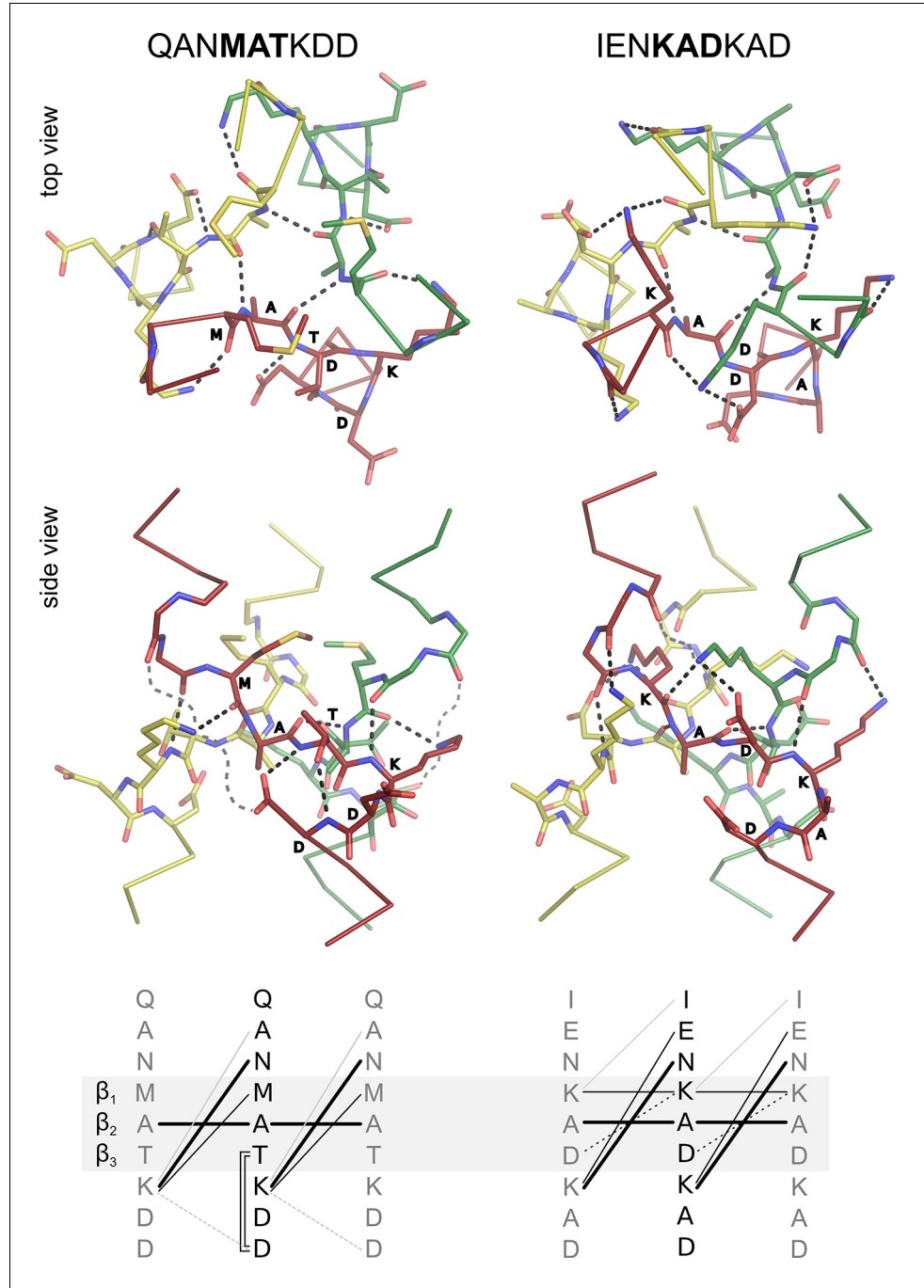

**Figure 8.** Interaction networks of canonical β-layers. The two distinct interaction networks of N-capping β-layers based on the sequence MATKDD and C-capping β-layers based on the sequence KADKAD are compared. The upper panels show the interactions at the first β-layer in the Tcar0761 structure and the β-layer in the A9 structure, both in top and side view. For clarity, the side views show only the interactions of the red chain. The lower panels show a schematic representation of the interactions: invariant backbone-to-backbone hydrogen bonds are drawn as bold lines, network-specific backbone-to-sidechain and sidechain-to-sidechain interactions are drawn as solid and broken lines, respectively. Grey lines indicate alternative/additional interactions which are not formed in the depicted β-layers but can be found in other instances as described in the main text. These interactions are indicated by loose grey broken lines in the side views.

capping interaction. However, when the $\beta_1$ residue is K (mainly in the stalks of TAAs and related phage proteins) the interaction network changes entirely from an N-cap of the following helices to a C-cap of the preceding ones. The K in $\beta_1$ reaches across the core of the trimer to form one, two, or all three of the following interactions: coordinate the backbone carbonyls of the $\alpha_1$ and $\beta_1$ residues, and the sidechain of the $\beta_3$ residue, all from the neighboring chain (clockwise). Additionally, the K or R in position $e$ is entirely found in the C-capping conformation. In all cases, the network is completed by backbone hydrogen bonds from the $\alpha_3$ residues to the residues in $e$ (clockwise), as already described for OMP100 (*Figure 2C*). These considerations suggest that in a tandem of $\beta$-layers with hexad spacing, the first layer should favor a C-capping network, with K in position $\beta_1$, and the second an N-capping network, with a hydrophobic residue in $\beta_1$. This is in fact observed in the *Streptococcus* prophage tail-fiber protein (2C3F).

## Conclusions

The range of periodicities that α-helical coiled coils can assume is limited by the strain they impose on the constituent helices, as they progressively deviate from the 3.63 residues per turn of an undistorted α-helix. Insertions of 3 residues into a heptad background (stammers, 10/3 = 3.33) lead to the largest strain observed so far in continuous coiled coils and are accommodated by the local distortion of the α-helix into a $3_{10}$ helix. We find that increasing the strain further by insertions of 2x3 or 3x3 residues leads to a complete loss of helical structure and the local formation of short β-strands. These cross to form a triangular plane, which moves the path of each chain by 120° counterclockwise around the trimer axis. Within this plane, the central residues of the three β-strands form backbone hydrogen bonds whose geometry deviates substantially from that seen in β-sheets. We have named them β-layers and show that they can be brought about in a straightforward way by the insertion of 6 or 2 (9 = 2 modulo 7) residues into a heptad background. We propose that β-layers offer two clear advantages to protein fibers. They increase their resilience by tightly interleaving the monomers within the fiber and they offer a simple mechanism to integrate β-stranded domains into these fibers, thus increasing their functional complexity. Our results show that a novel backbone structure is accessible to the 20 proteinogenic amino acids in the allowed regions of the Ramachandran plot with only minor mutations to a known fold.

## Materials and methods

### Cloning

If not otherwise indicated, constructs were amplified by primer extension. Primers used for amplification, cloning and mutagenesis are listed in *Table 3*.

The OMP100 construct encompasses residues 133–198 of OMP100 from *Actinobacillus actinomycetemcomitans* (Genbank BAB86905.1), fused at both the N- and the C-terminus in heptad register to the trimeric leucine zipper GCN4-pII. The amplified DNA fragment was cloned in Eco31I-sites of pIBA-GCN4tri-His (*Hernandez Alvarez et al., 2008*).

The Tcar0761 construct is derived from open reading frame 0761 of *Thermosinus carboxydivorans* Nor1 (Genbank ZP_01667343.1). A DNA fragment encoding residues 68–101, fused directly to a fragment encoding 191–211, was made by gene synthesis (GenScript) and cloned in the Eco31I-sites of pIBA-GCN4 N16V-His.

The GCN4 N16V version of the pIBA-GCN4 series allows for the expression of protein fragments fused at both termini to GCN4 adaptors carrying the N16V mutation, a variant of the leucine zipper that forms a mixture of dimers and trimers. pIBA-GCN4 N16V-His was constructed by replacing the XbaI/HindIII fragment of pASK IBA2 by a DNA fragment containing the XbaI site, ribosomal binding site, N-terminal GCN4 N16V adaptor, multiple cloning site, C-terminal GCN4 N16V adaptor, (His)6-tag and the HindIII site. Aspartate residues in position $f$ of the first heptad were replaced by methionine and tryptophan in the N- and C-terminal GCN4 adaptor as described before (*Deiss et al., 2014*).

Constructs A6, T6 and T9 were amplified and cloned into the NdeI and BamHI sites of the expression vector pETHis1a_Nde1, a modified version of pETHis1a (*Bogomolovas et al., 2009*) allowing for expression of the constructs with a C-terminal (His)6-tag and a TEV-protease cleavage site. A7

**Table 3.** Primers used in this study.

| Construct | Primer |
|---|---|
| OMP100 | P omp1: 5'-GACCATGGTCTCCGATTCAGAACGTGCGCAGCACCGAAAACGCGGCGGCAGCCGCGGAACGAACAG<br>P omp2: 5'-GCTTTATCCGCTTTGTTTTCAATCGCTTTTGTTTTCCGCAATTTCTGTTCGTTCGCGCGGCTGC<br>P omp3: 5'-GAAAACAAAGCGGATAAAGCGGATGTGGAAAAAAACCGCGCGGATATTGCGGCGAACAGCCGCGCGATTGCGACCTTTCG<br>P omp4: 5'-GACCATGGTCTCCTCATTTTGGTGGTCAGCGCCGCAATGTTCTGGCTGCTGCGAAAGGTCGCAATCGGCGCG |
| pASK IBA GCN4 N16V | P iba1: 5'-ACAAAAATCTAGATAACGAGGGCAAAAAATGAAACAGCTGGAAATGAAAGTTGAAGAACTGCTGTCCAAAGTCTACCACCTGGAAAACGA<br>P iba2: 5'-CTCGAGGGATCCCCGGGTACCGAGCTCGAATTCGGGACCATGGTCTCCCAGTTTTTCAGACGCGCAACTTCGTTTTCCAGGTGGTAGAC<br>P iba3: 5'-GTACCCGGGGATCCCTCGAGAGGGGACCATGTCTCAATGAAACAGCTGGAATGAAAGTTGAAGAACTGCTGTCCAAAGTCTACCACC<br>P iba4: 5'-CACAGGTCAAGCTTATTAGTGATGGTGATGGCCAGAACCAACCAGTTTTTCAGACGCGCAACTTCGTTTTCCAGGTGGTAGACTTTGGACAGC |
| T6 | T6 p1: 5'-GGAATTCCATATGAAGCAGCTGGAAGACAAGGTGGAGGAACTGTGTCCAAAGTGTACCATCTCGAAAACGAGGTGGCGGCGTCTGAAGAAG<br>T6 p2: 5'-CTTGGACAGCAGTTCTTCCACCTTATCTTCCAGCTGCTTCAATCATCTTTGGTCGCCATCAGCTTCTTCAGACGCGCCACCTC<br>T6 p3: 5'-GGTGGAAGAACTGCTGTCCAAGGTGTATCATCTGGAGAATGAGTGGCGCGTCTGAAGAAGCTGGTGGGCGAACGCTGAGGATCCCG<br>T6 p4: 5'-CGGGATCCTCAGCGTTCGCCCACCAGCTTCTTCAGACGCGCCACTCATTCTCCAGATGATACACCTTGGACAGCAGTTCTTCCACC |
| T9 | T9 p1: 5'-GGAATTCCATATGAAGCAGCTGGAAGATAAGGTGGAAGAGCTGCTGTCAAAGTGTACCATCTGGAAAACGAAGTGGCGGCGTCTGAAGAAG<br>T9 p2: 5'-CAGCAGTTCTTCCACCTTATCTTCCAGCTGCTTCATGTTCGCCAATGTCATCTTTGGTCGCCATCAGCTTCTTCAGACGCGCCACTTC<br>T9 p3: 5'-GATAAGGTGGAAGAACTGCTGTCCAAAGTGTACCATCTGGAAAACGAAGTGGCGCGTCTGAAGAAACTGGTGGGCGAACGCTGAGGATCCCG<br>T9 p4: 5'-CGGGATCCTCAGCGTTCGCCCACCAGTTTCTTCAGACGCGCCACTTCGTTTTCCAGATGGTACACTTGGACAGCAGTTCTTCCACCTTATC |
| A6 | A6 p1: 5'-GGAATTCCATATGAAGCAACTTGAAGACAAAGTCGAAGAGCTTCTCTCAAGTTTATCATCTTGAGAACGAAGTTGCTCGTCTTAAG<br>A6 p2: 5'-CCTTAGAAAGAAGTTCTTCGACCTTATCCTCAAGTTGCTTCATATCGCTTTGTCTCAATGAGTTTCTTAAGACGAGCAACTTCG<br>A6 p3: 5'-CGAAGAACTTCTTTCTAAGGTTACCATCTCGAAAATGAGGTTGTCGTTCAGAAGCTTGTTGGCGAACGCTGAGGATCCCG<br>A6 p4: 5'-CGGGATCCTCAGCGTTCGCCAACAAGCTTCTTGAGACGAGCAACCCATTTCGAGATGGTAAACCTTAGAAAGAAGTTCTTCG |
| A7 | MP A6+K se: 5'-CTTAAGAAAACTCATTGAGAACAAGAAAGCCGATATGAAGCAAC<br>MP A6+K as: 5'-GTTGCTTCATATCGGCTTTCTTGTTCTCAATGAGTTTCTTAAG |
| A9 | MP A6+KAD se: 5'-CATTGAGAACAAAGCCGATAAGGCTGACATGAAGCAACTTGAGG<br>MP A6+KAD as: 5'-CCTCAAGTTGCTTCATGTCAGCCTTATCGGCTTTGTTCTCAATG |

and A9 were constructed by site-directed mutagenesis using DNA fragment A6 as a template following the instructions of the QuikChange II XL Site-Directed Mutagenesis Kit. The DNA fragment coding for variant A9b was produced by gene synthesis (GenScript) and cloned in the NdeI and BamHI sites of pETHis1a_Nde1.

### Protein expression and purification

A6, A7, A9, A9b, T6 and T9 were expressed in *E. coli* strain C41 (DE3), OMP100 and Tcar0761 constructs in XL1-blue. Cells were grown at 37°C until OD600 = 0.6, then expression was induced by addition of 1 mM isopropyl β-D-1-thiogalactopyranoside. Cells were cultivated for another 5 hr, harvested by centrifugation and disrupted using a French press cell (SLM Aminco). All proteins were purified under denaturing conditions. 6 M guanidinium chloride was added to the cell lysate and the sample stirred for 1 hr at room temperature. After centrifugation, the supernatant was loaded on a NiNTA column equilibrated with 20 mM Tris, pH 7.9, 400 mM NaCl, 10% glycerol, 6 M guanidinium chloride and bound proteins were eluted with a linear gradient of 0–0.5 M imidazol. Proteins were refolded by dialysis. Corresponding refolding buffers are listed in *Table 1*. Refolded OMP100 was additionally subjected to a Superdex 75 column. For A6, A7, A9, A9b, T6 and T9 the N-terminal histidine tags were removed before crystallization. As the TEV cleavage site turned out to be not accessible for the TEV protease, the N-terminal tag was digested with Proteinase K. Subsequent analysis of the proteins by mass spectroscopy showed intact proteins lacking only the N-terminal extension including the histidine tag and the TEV cleavage site.

### X-ray crystallography and structure analysis

Crystallization trials were set up in 96-well sitting-drop plates with drops consisting of 400 nl protein solution + 400 nl reservoir solution (RS) and reservoirs containing 75 µl RS. Crystallization and cryo-protection conditions for all crystal structures are listed in *Table 4*. All crystals were loop mounted, flash frozen in liquid nitrogen, and all data collected at the SLS (Paul Scherrer Institute, Villigen,

**Table 4.** Crystallization and cryo condition.

| Structure | Protein solution & concentration | Reservoir solution (RS) | Cryo solution |
|---|---|---|---|
| OMP 100 | 20 mM Tris pH 7.5, 150 mM NaCl, 3% (v/v) Glycerol, 3 mg/ml protein | 0.1 M tri-Sodium citrate pH 5.5, 2% (v/v) Dioxane 15% (w/v) PEG 10,000 | RS + 15% (v/v) PEG 400 |
| A6 | 20 mM HEPES pH 7.2, 50 mM NaCl, 2% (v/v) Glycerol, 1 M Urea, 15 mg/ml protein | 95 mM tri-Sodium citrate pH 5.6, 19% (v/v) Isopropanol, 19% (w/v) PEG 4000, 5% (v/v) Glycerol | - |
| A7 | 20 mM HEPES pH 7.3, 50 mM NaCl, 1 M Urea, 15 mg/ml protein | 0.1 M Citric acid pH 3.5, 3 M NaCl | - |
| A9 | 20 mM HEPES pH 7.2, 50 mM NaCl, 2% (v/v) Glycerol, 1,5 M Urea, 17 mg/ml protein | 1.6 M tri-Sodium citrate pH 6.5 | - |
| A9b black | 50 mM HEPES, 50 mM NaCl, 1 M Urea, 7.5 mg/ml protein | 2.4 M Sodium malonate pH 5.0 | - |
| A9b grey | 50 mM HEPES, 50 mM NaCl, 1 M Urea, 7.5 mg/ml protein | 0.2 M Sodium citrate, 0.1 M Bis Tris propane pH 6.5, 20% (w/v) PEG 3350 | - |
| T6 | 20 mM HEPES pH 7.2, 50 mM NaCl, 1 M Urea, 13 mg/ml protein | 0.2 M $CaCl_2$, 0.1 M HEPES pH 7.5, 30% (w/v) PEG 4000 | - |
| T9$_6$ | 20 mM HEPES pH 7.2, 50 mM NaCl, 2% (v/v) Glycerol, 1.5 M Urea, 15 mg/ml protein | 0.2 M Ammonium phosphate, 0.1 M TRIS pH 8.5, 50% (v/v) MPD | - |
| T9$_9$ | „ | 0.1 M Citric acid pH 5.0, 20% (v/v) Isopropanol | RS + 1 M Urea +25% Glycerol |
| Tcar 0761 | 20 mM MOPS pH 7.2, 400 mM NaCl, 5% (v/v) Glycerol, 1.5 M Urea, 7 mg/ml protein | 0.1 M tri-Sodium citrate pH 4.0, 30% (v/v) MPD | - |

**Table 5.** Data collection and refinement statistics.

| Structure | OMP100 | A6 | A7 | A9 | A9b black | A9b grey | T6 | T9$_6$ | T9$_9$ | Tcar0761 |
|---|---|---|---|---|---|---|---|---|---|---|
| Beamline/Detector* | PXII / M | PXII / M | PXII / M | PXIII / M | PXII / P | PXII / P | PXII / P | PXII / M | PXIII / M | PXII / P |
| Wavelength (Å) | 0.9786 | 1.0 | 1.0 | 1.0 | 1.0 | 1.0 | 1.0 | 1.0 | 1.0 | 1.0 |
| Trimers/AU | 1 | 1 | 1/3 | 1 | 1 | 2 | 1 | 1 | 1 | 1/3 |
| Space group | C2** | C2** | P321 | P2$_1$ | P2$_1$ | P2$_1$ | P2$_1$ | P2$_1$ | C2** | P6$_3$ |
| a (Å) | 62.1 | 60.4 | 38.2 | 65.2 | 26.2 | 71.1 | 34.2 | 25.1 | 60.8 | 37.9 |
| b (Å) | 35.9 | 34.8 | 38.2 | 34.6 | 37.5 | 35.0 | 27.0 | 38.3 | 35.1 | 37.9 |
| c (Å) | 198.5 | 104.2 | 87.1 | 67.5 | 95.0 | 106.2 | 101.0 | 105.0 | 112.2 | 179.2 |
| β (°) | 96.0 | 101.1 | 90 | 117.7 | 92.6 | 101.7 | 93.9 | 93.3 | 100.4 | 90 |
| Resolution range (Å)*** | 32.9–2.30 (2.44–2.30) | 30.0–2.10 (2.23–2.10) | 18.2–1.37 (1.45–1.37) | 33.7–1.80 (1.91–1.80) | 34.9–1.35 (1.43–1.35) | 38.1–2.00 (2.12–2.00) | 34.1–1.60 (1.70–1.60) | 34.9–1.80 (1.91–1.80) | 19.5–2.00 (2.12–2.00) | 32.3–1.60 (1.69–1.60) |
| Completeness (%) | 92.4 (86.5) | 97.3 (96.2) | 99.0 (98.6) | 98.9 (97.4) | 95.9 (92.1) | 92.4 (98.9) | 98.2 (96.1) | 97.1 (95.4) | 98.7 (97.5) | 99.2 (96.9) |
| Redundancy | 2.84 (2.52) | 3.71 (3.71) | 6.35 (6.33) | 3.70 (3.67) | 3.72 (3.47) | 3.29 (3.31) | 3.04 (2.89) | 3.94 (3.81) | 3.73 (3.73) | 3.69 (3.65) |
| I/σ(I) | 14.0 (1.88) | 15.5 (2.28) | 18.2 (2.52) | 14.3 (2.07) | 17.6 (2.10) | 13.9 (2.43) | 13.6 (2.33) | 14.5 (2.14) | 19.5 (2.25) | 20.3 (2.23) |
| R$_{merge}$ (%) | 4.2 (44.8) | 4.8 (62.1) | 5.1 (75.5) | 5.1 (61.7) | 3.4 (66.6) | 5.0 (51.5) | 4.4 (42.3) | 7.2 (71.7) | 4.0 (63.2) | 2.9 (60.2) |
| R$_{cryst}$ (%) | 22.5 | 20.8 | 19.5 | 20.6 | 16.3 | 20.6 | 17.4 | 18.7 | 21.1 | 17.7 |
| R$_{free}$ (%) | 25.4 | 25.1 | 23.8 | 25.6 | 19.9 | 25.3 | 20.5 | 22.6 | 25.5 | 21.3 |
| PDB code | 5APP | 5APQ | 5APS | 5APT | 5APU | 5APV | 5APW | 5APX | 5APY | 5APZ |

*M = MARRESEARCH mar225 CCD detector; P = DECTRIS PILATUS 6M detector

**twinned with apparent H32 symmetry and twinning operators

1/2*h-3/2*k,-1/2*h-1/2*k,-1/2*h+1/2*k,-l and 1/2*h+3/2*k,1/2*h-1/2*k,-1/2*h-1/2*k-l

***values in parenthesis refer to the highest resolution shell

Switzerland) under cryo conditions at 100 K using the beamlines and detectors indicated in *Table 5*. Data were processed and scaled using XDS (*Kabsch, 1993*). Structures were solved by molecular replacement using MOLREP (*Vagin and Teplyakov, 2000*). For OMP100 and the GCN4-fusion constructs, trimmed models of the SadAK3 structure (2WPQ) were used as search models. For Tcar0761, fragments of the T6 and A9 structures were used. After rebuilding with ARP/WARP (*Perrakis et al., 1999*), all structures were completed in cyclic manual modeling with Coot (*Emsley and Cowtan, 2004*) and refinement with REFMAC5 (*Murshudov et al., 1999*). Analysis with Procheck (*Laskowski et al., 1993*) showed excellent geometries for all structures. Data collection and refinement statistics are summarized in *Table 5*. Periodicity plots were calculated based on the output of TWISTER (*Strelkov and Burkhard, 2002*). Molecular depictions were prepared using MolScript (*Kraulis, 1991*), Raster3D (*Merritt and Bacon, 1997*) and Pymol (Schrödinger, LLC, New York, NY).

## Bioinformatics

Sequence similarity searches were carried out at the National Institute for Biotechnology Information (NCBI; http://blast.ncbi.nlm.nih.gov/) and in the MPI Bioinformatics Toolkit (http://toolkit.tuebingen.mpg.de; *Biegert et al., 2006*), using PSI-Blast (*Altschul et al., 1997*) at NCBI and PatternSearch, CS-Blast (*Biegert and Söding, 2009*), HMMER (*Eddy, 2011*), HHblits (*Remmert et al., 2011*) and HHpred (*Söding et al., 2005*) in the MPI Toolkit. The sequence relationships of proteins identified in these searches were explored by clustering them according their pairwise Blast P-values in CLANS (*Frickey and Lupas, 2004*). Sequence logos were created from representative, non-redundant alignments using the WebLogo3 web server (*Crooks et al., 2004*) with composition correction switched off.

Secondary structure propensity was evaluated in the MPI Toolkit with the meta-tools Quick2D and Ali2D, and coiled-coil propensity was estimated with COILS/PCOILS (*Lupas et al., 1991*; *Gruber et al., 2006*) and MARCOIL (*Delorenzi and Speed, 2002*).

Searches for structures containing $\beta$-layers were performed over the Protein Data Bank (PDB, Dec 8 2015) in a two-step procedure: First, their torsion angles were scanned with seven-residue sliding windows of $\beta\beta\beta\alpha\alpha\alpha\alpha$ and $\alpha\alpha\alpha\alpha\beta\beta\beta$, where $\alpha$ must satisfy $-70° \leq \psi \leq -10°$ and $-180° \leq \varphi \leq -40°$, and $\beta$ must satisfy $20° \leq \psi \leq 180°$ and $-180° \leq \varphi \leq -40°$. Second, the central $\beta$ residue of putative $\beta$-layer strands was required to form backbone hydrogen bonds (N-O distance $\leq 3.5$ Å) to the equivalent residue of another $\beta$-layer strand within a biological assembly. All matches were verified by visual inspection. These searches were complemented by extensive interactive analyses of fibrous proteins in PDB.

## Acknowledgements

We thank Reinhard Albrecht, Kerstin Baer and Silvia Deiss for technical assistance and are very grateful to the staff of beamline X10SA/Swiss Light Source for excellent technical support. This work was supported by the German Science Foundation (SFB 766, TP B4) and by institutional funds of the Max Planck Society.

## Additional information

### Funding

| Funder | Grant reference number | Author |
|---|---|---|
| Max-Planck-Gesellschaft | | Marcus D Hartmann<br>Claudia T Mendler<br>Jens Bassler<br>Ioanna Karamichali<br>Oswin Ridderbusch<br>Andrei N Lupas<br>Birte Hernandez Alvarez |
| Deutsche Forschungsgemeinschaft | SFB766 (TP B4) | Andrei N Lupas<br>Birte Hernandez Alvarez |

The funders had no role in study design, data collection and interpretation, or the decision to submit the work for publication.

## Author contributions
MDH, BHA, Conception and design, Acquisition of data, Analysis and interpretation of data, Drafting or revising the article; CTM, JB, IK, OR, Acquisition of data, Analysis and interpretation of data; ANL, Conception and design, Analysis and interpretation of data, Drafting or revising the article

## Author ORCIDs
Marcus D Hartmann, http://orcid.org/0000-0001-6937-5677
Andrei N Lupas, http://orcid.org/0000-0002-1959-4836

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
