## [Decision Letter]

Thank you for submitting your work entitled "α/β Coiled Coils" for consideration by *eLife*. Your article has been favorably evaluated by John Kuriyan (Senior editor) and three reviewers, one of whom, Mingjie Zhang, is a member of our Board of Reviewing Editors. The other two reviewers have also agreed to reveal their identity: David Baker and William DeGrado (peer reviewers).

The reviewers have discussed the reviews with one another and the Reviewing editor has drafted this decision to help you prepare a revised submission.

Summary:

In this manuscript, the authors systematically studied the structural impact of insertion of 2 or 6 residues into classical heptad repeat sequences on formation of coiled coil structures. The study builds on a previous finding of a so-called β-layer structure found in *Salmonella enterica* autotransporter adhesin SadA from the same group several years ago (Hartmann et al., PNAS, 2012). Via working with a series of natively occurring bacterial adhesion fragments or designed derivatives, the authors showed that formation of the β-layer structure is a common strategy for the proteins to "solve" energetic challenges imposed by the insertion. The work uncovers a new coiled-coil assembly mode and offers a mechanistic understanding for such assembly. The work will be of great interest to scientists working on coiled coil design as it provides a route to expanding the repertoire of accessible structures. The described α/β coiled coils may also serve as templates for designing various protein assemblies.

Essential revisions:

1) The most critical issue is perhaps the minimal length required to form a stable α/β coiled coil. In all of the constructs described in the study, the two ends of the proteins are "protected" by highly stable GCN4 coiled coils. In addition, the authors found that in certain designed sequences (e.g. the A9 construct), the residues form the β-layer structure are different in different crystals. This naturally leads to the issue of which amino acid residues can or cannot form the β-layer structure. It will be very useful to have some biochemical information on this aspect. Such information would also be needed to support the authors' claim the formation of the β-layer can "increase their resilience" for coiled coil protein fibers.

2) It is not clear in the manuscript whether formation of the β-layer structure can only occur for trimeric, parallel coiled coils. This is an important issue, as it is still difficult to predict whether a protein sequence with obvious coiled-coil features will form a trimer or not. If the β-layer structure can only form trimer, would it be possible to combine the insertion sequence to improve the prediction score for trimeric coiled coils?

3) To address how widespread the β layer is and its possible roles in biology, could the authors provide a list of proteins predicted based on insertion lengths and positions to contain β layers, and assess whether there are previously unnoticed β-layers already in the PDB? What are the features for the α/β coiled coil proteins?

4) The overall figure presentations need to be improved. It is very hard to follow the figures. In essentially all figures, multiple panels are stuffed in one figure without clear legend to guide reading. Figures are often not properly labeled to enhance the scientific clarity.

---

## [Author Response]

*1) The most critical issue is perhaps the minimal length required to form a stable α/β coiled coil. In all of the constructs described in the study, the two ends of the proteins are "protected" by highly stable GCN4 coiled coils. In addition, the authors found that in certain designed sequences (e.g. the A9 construct), the residues form the β-layer structure are different in different crystals. This naturally leads to the issue of which amino acid residues can or cannot form the β-layer structure. It will be very useful to have some biochemical information on this aspect. Such information would also be needed to support the authors' claim the formation of the β-layer can "increase their resilience" for coiled coil protein fibers.*

Excellent points. We did wonder all along the project whether β-layers stabilized the coiled coils, destabilized them, or were neutral in this respect, but the biophysical behavior of our various constructs did not follow recognizable patterns.

We note however that β-layers can form stably at the N-terminal end of coiled-coil proteins, as N-caps to the coiled coil, without N-terminal “protectors”. That is clearly seen in the structure of the fusion-pH form of influenza hemagglutinin HA2, where this structure has in fact been noted and described by Wiley and colleagues, as well as in two other structures currently in PDB (SadB, MPN010). We now state this explicitly in the revised text.

β-Layers can also occur at the C-terminal ends of coiled coils, close to the ends of the protein, but not as clearly “unprotected” as the N caps.

This said, we used the term “resilience” in the paper to indicate the ability to recover the native structure after local denaturation, for instance after exposure to harsh conditions, not in order to make any claim about stability, which we do not think we can make. This view is supported by the high incidence of β-layers in trimeric adhesins and phage envelope proteins, which we note in a new Table 2; both are surface proteins optimized to withstand a wide range of environmental conditions. We think that resilience is a corollary of the intertwining of the chains that results from β-layers, keeping the chains together even after partial denaturation and allowing for a more efficient recovery of the native structure.

We address the issue of residues compatible with a β-layer structure in a detailed new section of the revised paper and in a new Figure 6.

*2) It is not clear in the manuscript whether formation of the β-layer structure can only occur for trimeric, parallel coiled coils. This is an important issue, as it is still difficult to predict whether a protein sequence with obvious coiled-coil features will form a trimer or not. If the β-layer structure can only form trimer, would it be possible to combine the insertion sequence to improve the prediction score for trimeric coiled coils?*

We do not think that this structure could be formed in any other type of coiled coil, due to the specific geometry of the β interactions in the layer. In revision we searched for β-layers of known structure in PDB using backbone torsion angles in a fairly generous range and the requirement that the middle residue form a backbone hydrogen bond to a symmetry-related subunit, without specifying the symmetry. We still only got trimers back (as now listed in the new Table 2, with 31 entries).

The detection of β-layers would certainly be diagnostic of parallel, trimeric coiled coils, but we are still struggling to predict them from sequence data at present.

*3) To address how widespread the β layer is and its possible roles in biology, could the authors provide a list of proteins predicted based on insertion lengths and positions to contain β layers, and assess whether there are previously unnoticed* β

*-layers already in the PDB? What are the features for the α/β coiled coil proteins?*

We have now added an entire new section to the paper (>1200 words, three figures, one table), in which we describe the protein families we have found so far to contain β-layers by sequence searches, as well as the proteins in PDB that we found to contain β-layers by structure searches. We interpret the consensus sequences in light of the structural data and show that β-layers are most often found as N cap structures at the N-terminus of coiled coils, less frequently at the C-terminus of coiled coils and least frequently as structural elements within coiled coils. In the latter context, they can either assume an N-cap or a C-cap role, depending on their sequence properties.

4) The overall figure presentations need to be improved. It is very hard to follow the figures. In essentially all figures, multiple panels are stuffed in one figure without clear legend to guide reading. Figures are often not properly labeled to enhance the scientific clarity.

We have revised all figures, particularly Figure 1 and Figure 3, to simplify the number of panels, adjust the position of the labels, and coordinate the color scheme between figures. We have also revised the figure legends.